# Quality Assessment of Processed Graphene Chips for Biosensor Application

**DOI:** 10.3390/ma16165628

**Published:** 2023-08-15

**Authors:** Natalia M. Shmidt, Evgeniya I. Shabunina, Ekaterina V. Gushchina, Vasiliy N. Petrov, Ilya A. Eliseyev, Sergey P. Lebedev, Sergei Iu. Priobrazhenskii, Elena M. Tanklevskaya, Mikhail V. Puzyk, Alexander D. Roenkov, Alexander S. Usikov, Alexander A. Lebedev

**Affiliations:** 1Ioffe Institute, 26 Politekhnicheskaya, 194021 St. Petersburg, Russia; natalia.shmidt@mail.ioffe.ru (N.M.S.); jenni-85@mail.ru (E.I.S.); katgushch@yandex.ru (E.V.G.); krishkis@i.ua (V.N.P.); zoid95@yandex.ru (I.A.E.); lebedev.sergey@mail.ioffe.ru (S.P.L.); sereyozha@yandex.ru (S.I.P.); elena.tanklevskaya@gmail.com (E.M.T.); 2Faculty of Chemistry, Herzen State Pedagogical University of Russia, 191186 St. Petersburg, Russia; puzyk@mail.ru; 3Nitride Crystals Group, 194156 St. Petersburg, Russia; roenkov47@yandex.ru (A.D.R.); alexander.usikov@nitride-crystals.com (A.S.U.)

**Keywords:** graphene chip, surface topography, photoresist residues, low-frequency noise

## Abstract

The quality of graphene intended for use in biosensors was assessed on manufactured chips using a set of methods including atomic force microscopy (AFM), Raman spectroscopy, and low-frequency noise investigation. It is shown that local areas of residues on the graphene surface, formed as a result of the interaction of graphene with a photoresist at the initial stage of chip development, led to a spread of chip resistance (R) in the range of 1–10 kOhm and to an increase in the root mean square (RMS) roughness up to 10 times, which can significantly worsen the reproducibility of the parameters of graphene chips for biosensor applications. It was observed that the control of the photoresist residues after photolithography (PLG) using AFM and subsequent additional cleaning reduced the spread of R values in chips to 1–1.6 kOhm and obtained an RMS roughness similar to the roughness in the graphene film before PLG. Monitoring of the spectral density of low-frequency voltage fluctuation (S_U_), which provides integral information about the system of defects and quality of the material, makes it possible to identify chips with low graphene quality and with inhomogeneously distributed areas of compressive stresses by the type of frequency dependence S_U_(f).

## 1. Introduction

Graphene is a single or few atoms thick sheet of sp^2^-bonded carbon atoms in a closely packed honeycomb 2D lattice. Unique properties of graphene such as large surface area, high adsorption capacity and mobility, low thermal and 1/f noise characteristics, high elasticity, good charge-transfer ability, and ferromagnetic properties are well known [1]. Thus, graphene has gained wide popularity in chemical sensing applications, electronics, biomedical applications, including biosensing, medical imaging, drug/gene delivery, photothermal therapy, tissue engineering, and immunotherapy [2,3,4,5,6,7,8,9,10,11].

The graphene industry has been developing rapidly in recent years. However, a number of issues remain which need to be addressed to improve technological progress of the graphene-based biosensors [12]. Firstly, the repeatability and constancy of graphene properties, such as size, thickness, and the number of functional groups, differ despite similar synthesis and functionalization methods. Secondly, the performance characteristics of graphene devices are often limited by deformation, defects, and impurities induced during device fabrication.

It should be noted that the largest number of stages of graphene processing is typical of biosensor fabrication, including viral biosensors [3,4,5,6,7,8,9,10]. The concept of a graphene-based biosensor is based on the antigen-antibody immunoreaction on the graphene surface.

Briefly, the main stages of creating biosensors include the formation of a graphene film, the formation of chips with contact pads (graphene resistors or transistor) by photolithography (PLG), controlled treatment (functionalization) of the graphene surface in chips, immobilization (attachment) of biomolecules (antibodies), and the implementation of an antibody–antigen reaction [3,4,5,6,7,8,9]. Such a sophisticated biosensor fabrication technique is necessary to increase the selectivity and sensitivity of biosensors. Its use has made it possible to obtain biosensors for viral infection, such as A and B influenza and COVID-19 [3,5,11,12,13,14,15,16]. The biosensor could detect the SARS-CoV-2 spike protein at a concentration of 1.3 × 10^−5^ pM [16]. However, these successes do not yet allow for the transition to the industrial production of viral biosensors and their widespread use in medicine [16]. It is necessary to improve the stability and reproducibility of parameters, increase the selectivity of biosensors, and develop standards for the industrial quality of graphene and its control [16].

As was shown earlier, the non-reproducibility of graphene properties in chips and films can occur at any stage of chip manufacturing [11,12]. The quality of graphene films can already be disturbed at the initial stage of the chip formation of a given topology by photolithography (PLG) [11,17,18,19].

It is shown that the photoresist interacts with graphene creating local regions with resist residues (LRRs) not completely removed from the graphene surface [17,18,19,20,21]. This problem cannot be fully solved by choosing a specific photoresist, since the interaction takes place at the level of benzene rings, which are present in all types of photoresists used, and in graphene (regardless of the method of its production) [18]. The traditional method of monitoring the photoresist removal process in an optical microscope does not allow detection of LRRs on graphene.

The presence of LRRs impairs graphene morphology, enhances the heterogeneous deformation distribution and non-reproducibility of biosensor parameters even before graphene functionalization, antibody immobilization, and antigen (virus) detection [11,17,18,19]. In particular, it was shown that the resistance value of graphene chips (biosensors) obtained from the same film can vary from 2 to 10 times. The reproducible resistance values are observed for only 20% of biosensors [11].

One of the methods to resolve the problem is to use a protective (sacrificial) layer between the graphene film and a photoresist to limit the interaction between them [19,20,21]. It should be noted that it is quite difficult to compare the effectiveness of the proposed methods based on published data. The problem relates to the variety of graphene properties obtained by different methods and the lack of generally accepted graphene quality standards. Since 2014 [19] and up to the present, work on the successful solution of the problem has been regularly published [11,21], yet there does not seem to be a generally accepted, effective way to solve the problem. It should be noted that several papers provide data that the introduction of a protective (sacrificial) layer was insufficient. It was possible to obtain a graphene surface free of LRRs only with additional annealing [19,20]. Thus, despite the successes achieved in cleaning the graphene surface, a generally accepted solution has not yet been found. Meanwhile, this is an important step in obtaining biosensors with reproducible parameters.

In this work, we report on the quality assessment of processed graphene chips, obtained by traditional PLG and PLG with additional LRRs cleaning, and on the reproducibility of their parameters for biosensor applications. Additional LRR cleaning is based on the use of piranha solution, which is widely used in microelectronics and in silicon technology to remove organic contaminants, but was not used in graphene PLG, according to our data. We assess the quality of graphene in chips using traditional AFM [10,22] and Raman [11,23,24,25] studies and low-frequency noise (LFN) studies. Raman spectroscopy is widely used to confirm graphene films and to characterize them. AFM is also used to study the morphology of graphene films and nm-scale residues on its surface.

It is very important that the LFN method [26] integrally characterizes the system of defects of the entire area between the contacts in the chip, not only the micro region as with AFM and Raman methods. Being a two-dimensional structure with widely tunable two-dimensional carrier concentration, monolayer graphene films offer unique opportunities for studying their 1/f noise [26]. The nature of this noise is diverse and reflects fluctuation processes in the material, such as fluctuations in carrier mobility, fluctuations in carrier concentration, fluctuations in barrier height, charge fluctuations in surface states, as well as the charge state of defects with different activation energies [26,27,28].

Fluctuations in population levels form a tail of the density of states near the boundaries of the conduction and the valence band. These fluctuations can be caused by various imperfections of the crystal lattice, including clusters, extended defects, local stresses, inhomogeneities of composition, and non-ohmic contacts.

An independent investigation of 1/f noise in a wide range of graphene devices (μ from 400 to 20,000 cm^2^·V^−1^·s^−1^) showed that in most of the studied devices the dominant contribution to 1/f noise was made by mobility fluctuations arising from fluctuations in the scattering cross-section σ [26,27].

Low-frequency 1/f noise caused by mobility fluctuations can occur as a result of the superposition of elementary events in which the scattering cross-section of the scattering centers σ fluctuates from σ_1_ to σ_2_. For graphene films, the dependence of the spectral density of a low-frequency current noise (S_I_) as well as that of a low-frequency voltage fluctuations (S_U_) on frequency have the form S_U_ (or S_I_) ~ 1/f^γ^ in the frequency range 1–100 Hz. The quality of graphene and the reproducibility of its properties in the chips in general can be judged by comparing the magnitude of low-frequency noise at a frequency of 1.22 Hz between the chips with the same geometry and from the same film. We discuss the impact of the identified reasons of non-reproducibility of graphene parameters in chips on the properties of chips for biosensor applications.

## 2. Materials and Methods

The graphene films were formed on semi-insulating 4H-SiC substrates by thermal decomposition of the (0001) Si surface in a graphite crucible with induction heating [29]. The method allows one to obtain high-quality graphene films on a high-resistance substrate of arbitrarily large areas, which is important to process graphene chips (dies) for sensing applications. Substrates were purchased from TANKEBLUE Co., Ltd., (Beijing, China). The growth temperature was 1730 ± 20 °C, the argon pressure in the growth chamber was 750 ± 20 Torr, and the growth time was 5 min. High-purity (99.9999%) argon (Centrgas Co., Ltd. Saint-Petersburg, Russia) was used in the growth process.

The ordinary photolithography process was applied to form a topological pattern of as-grown graphene/SiC films utilizing AZ1318 photoresist (Merck Performance Materials, Darmstadt, Germany). The photoresist was removed in acetone, and the graphene surface was examined by AFM followed by additional cleaning of the LRRs. Additional cleaning based on a piranha solution H_2_O_2_ (30%) + H_2_SO_4_ (concentrated) 1:2, (Neva Reaktiv Co., Ltd. Saint-Petersburg, Russia, https://nevareaktiv.ru/, accessed on 11 August 2022) at 25–27 °C was used. Details of the graphene films and the processing and mounting of chips on holders can be found elsewhere [29,30]. Chips with two contact pads (graphene resistors—the base of biosensors) are shown in Figure 1a,b. The size of the sensing area (active surface of graphene in the chip) is about 0.8 × 0.8 mm^2^.

The presence of graphene films on the SiC surface was confirmed, and their structure was characterized by Raman spectroscopy. Measurements were performed at room temperature in the backscattering geometry using a Horiba LabRAM HREvo UV-VIS-NIR-Open spectrometer (Horiba, Lille, France) equipped with a confocal microscope. A YAG:Nd laser (Torus, Laser Quantum, Stockport, UK) with a wavelength of 532 nm was used as an excitation source. The laser beam was focused on the area with ~1 µm diameter using an Olympus MPLN100x (Olympus, Tokyo, Japan) objective lens (NA = 0.9). The laser power nm was limited to 4.0 mW to prevent damaging and modification of graphene films.

The AFM measurements of the graphene surface morphology were carried out on the Ntegra AURA setup (NTMDT, Moscow, Zelenograd, Russia). The AFM studies were carried out using the HA_FM cantilever (www.tipsnano.com, accessed on 11 August 2022) in a resonant mode of operation. The AFM probe knocks on the surface scanning frequency 0.6 Hz. The stiffness coefficient of such a cantilever is 3.5 N/m, the radius of curvature is less than 10 nm, and the scanning field size is 256 × 256 points.

The power spectral density of voltage fluctuations (S_U_) was measured for the graphene chips mounted on a holder in a frequency range from 1 to 50 kHz. The studied chips were connected in series with a low-noise load resistor RL whose resistance varied from 100 Ohm to 13 kOhm depending on the current passing through the chip. The voltage fluctuations S_U_ at the resistors RL were amplified by a low-noise preamplifier SR560 (Stanford Research Systems, Sunnyvale, CA, USA) and subsequently measured by an SR 770 FET NETWORK Analyzer (Stanford Research Systems, Sunnyvale, CA, USA). The background noise of the preamplifier did not exceed 4 nV/√Hz at 1 kHz, which is approximately equivalent to the Johnson–Nyquist noise of a 1000-Ohm resistance. The I–U characteristics of the chip were measured using the KEITHLEY 6487 (Tektronix, Inc., Beaverton, OR, USA) power source.

## 3. Results and Discussion

To determine the reproducibility of graphene properties in chips processed from as-grown graphene/SiC films, the Raman spectra and surface topography of graphene were studied both in chips and in as-grown graphene/SiC films. The typical Raman spectra of the as-grown graphene/SiC film consisted of sharp G and 2D lines characteristic of monolayer graphene [31] and several bands centered at approximately 1230, 1380, and 1550 cm^−1^ corresponding to the buffer layer [11,25] as shown in Figure 1c. High structural quality of our as-grown graphene/SiC films is indicated by the absence of the defect-related D line. Surface topography of the as-grown graphene/SiC film is given in Figure 2a. The root mean square roughness (RMS) of as-grown graphene films was 0.45–0.55 nm at AFM scan 10 µm× 10 µm.

This study and comparison of the properties of graphene in chips were carried out on chips obtained from the same film.

It is known that the processing of graphene chips from as-grown graphene/SiC films using conventional PLG with the removal of a photoresist in acetone, leads to a deterioration in the morphology of the graphene surface and a set of LRRs [11,17,18]. Our experiments have shown that LRRs contribute to the non-reproducibility of chip parameters [11]. A comparison of the AFM images in Figure 2a,b reveals a typical change in the surface topography of graphene caused by the formation of LRRs visible as white local regions in AFM images. The presence of LRRs results in spread RMS values from 0.5 to 10 nm at an AFM scan of 10 µm × 10 µm in graphene chips from the same film. At the same time, there was a spread of resistance values (R) in chips from 1 to 10 kOhm, which is associated with the spread of RMS values. Large values of resistance in chips, as a rule, were accompanied by a large RMS value of graphene surface roughness. As a result, only 20% of the 50 chips, which had resistance values in the range of 1–1.6 kOhm and an RMS roughness from 0.5 to 1 nm with AFM scanning of 10 µm × 10 µm, can be considered as chips with nearly the same (uniform) parameters.

In this work, to remove LRRs identified by AFM, additional cleaning based on a piranha solution H_2_O_2_ (30%) + H_2_SO_4_ (concentrated) 1:2, (Neva Reaktiv Co., Ltd. Saint-Petersburg, Russia, https://nevareaktiv.ru/, accessed on 11 August 2022) was used. After the cleaning, 50 chips fabricated from the as-grown graphene/SiC film (EG417 series) were evaluated based on the AFM characterization, the results of resistance measurement, the level of spectral density of a low-frequency noise (S_U_) at a frequency of 1.22 Hz, as well as analyzing the shape of S_U_ frequency dependencies. No RMS values larger than 2 nm were observed in graphene in the vast majority of chips. The typical RMS values are about 0.60 nm (Figure 2c), which coincides with, or is close to, the initial RMS values of as-grown graphene/SiC film (RMS = 0.55 nm) Figure 2a. A significant reduction in the spread of resistance values between the chips to 1–1.6 kOhm was observed.

It is difficult to say that the method used here is better than the technique described earlier [19,21], which uses the introduction of a sacrificial layer (an additional photoresists) or thin metal films. There are no generally accepted standards; not only for the quality of the graphene surface after post-growth processes, but also for the as-grown graphene surface. That is why we compare the topography of the graphene surface and the RMS values after the conventional removal of the photoresist in acetone and after additional cleaning, as well as with the RMS values of graphene on the same as-grown film before PLG.

To find out the reasons for the resistance spread in the range of 1–1.6 kOhm, measurements of the spectral density of voltage fluctuations S_U_ at a frequency of 1.22 Hz were carried out for all 50 chips from the same film (EG417). We chose this technique because S_U_ values provide integral information about the state of the defect system of the whole graphene film in the chip, rather than some local areas. The histogram of the distribution of S_U_ values is shown in Figure 3a. For 82% of chips, the S_U_ values are as low as SU = (1 − 4) × 10^−13^ V^2^/Hz, which indicates a good quality graphene. Selective check-up of the Raman spectra for these chips shows that they are identical to those shown in Figure 1c.

For the remaining 18% of chips, the spread of S_U_ values was in the range of 6 × 10^−13^ to 8 × 10^−11^ V^2^/Hz (Figure 3a). Assuming that the observed variation of S_U_ in these chips can be caused by inhomogeneous deformations in graphene, the noise spectra S_U_(f) were investigated. It is known that the shape of noise spectra S_U_ ~1/f^γ^ (γ > 1) reveals the information about inhomogeneous deformations in the material [26,27]. Indeed, for the EG417-C5, E5, E7 chips showing different values of excess noise, there are regions with frequency dependencies S_U_ ~1/f^γ^ having different values of γ up to 1.4 with a maximum S_U_ value at 1.22 Hz for the EG417-C5 chip as shown in Figure 3b. Earlier, we have observed that the dependence of S_U_ ~1/fγ (γ > 1) in graphene films and the chips is consistent with the Raman spectra data that demonstrate an inhomogeneous distribution of compressive stresses in graphene [32]. Studying of graphene topography in these chips reveals cracks in graphene shown in Figure 4a–d (EG417-E7, -C5) and extended LRR regions shown in Figure 4e,g (EG417-E5) or a large number of heterogeneously spaced LRRs (Figure 2b and Figure 4e). We clarify that the formation of cracks is observed, as a rule, in processed graphene in chips after PLG. Several causes increase the likelihood of crack formation: boiling in acetone, elevated temperatures of photoresist hardening, prolonged exposure to ultrasound, and local deformations of graphene, as well as the ripple of as-grown graphene film.

As was shown previously [11], the heterogeneous distribution of deformations in the graphene chip used as a biosensor leads to non-uniform spread of the immobilized antibodies. The interaction of viruses (antigens) occurs not only to antibodies in accordance with antibody–antigen immunoreaction but on the part of the graphene surface that can give an error when determining the concentration of viruses by the biosensor. The formation of cracks in graphene leads to additional inhomogeneous deformations. Cracks also form uncontrollable areas to attach antigen directly to the graphene surface besides antibody–antigen immunoreaction. It destroys normal biosensor operation.

For 82% of the chips from the same film (EG417) with S_U_ values as low as S_U_ = (1 − 4) × 10^−13^ V^2^/Hz, the dependence S_U_ ~1/f^γ^ (γ > 1) is not observed. A typical dependence for these chips is illustrated by the EG417-D6 chip in Figure 3b. Two regions are clearly visible. In the frequency range 1–100 Hz, the experimental data fit to the dependence S_U_ ~1/f. A weaker dependence S_U_ ~1/f^0.8^ is observed for higher frequencies f >100 Hz. We believe the S_U_ dependence at a higher frequency may be due to a superposition of 1/f noise and generation-recombination (GR) noise. These features mean that, in addition to the system of defects typical to low-dimensional materials, there are single Shockley–Reed–Hall defects. It should be noted that this S_U_ behavior is observed in graphene in chips with the RMS values close to those on the as-grown graphene film before PLG (see Figure 2a). AFM and SEM images of small aggregates of SARS-CoV-2 viruses were obtained on chips with such characteristics [11]. The introduction of additional cleaning of the graphene surface in the chips also made it possible, as in [19,21], to preserve the quality of the graphene surface in chips after PLG with RMS values close to the RMS values of as-grown graphene film before PLG, to reduce the resistance of graphene in chips, and to increase the reproducibility of graphene resistance values in chips obtained from a single film, but in a simpler way than in [19,21]. The quality assessment of processed graphene in chips using the LFN method allowed us to control the heterogeneous distribution of deformations and isolate chips with cracked graphene that are not suitable for use in biosensors by the level of excessive noise. The use of this method in combination with AFM opens up opportunities to increase the reproducibility of the properties of processed graphene in chips, as well as biosensors, and to improve the technology for obtaining biosensors. The additional cleaning contributed to an increase in the sensitivity of viral biosensors and made it possible to successfully detect influenza B viruses (see more details in Appendix A and [11]).

## 4. Conclusions

The results of comparing the quality assessment of processed graphene in chips after conventional PLG and PLG with additional cleaning, as well as the reproducibility of the parameters of these chips, which are the basis for viral biosensors, are presented. Conventional PLG with the removal of the photoresist in acetone is accompanied by the formation of local regions with resist residues (LRRs). Significant non-reproducibility of graphene parameters in chips after conventional PLG is observed. The presence of LRRs on the surface of processed graphene result in a spread in RMS values from 0.5 to 10 nm at an AFM scan of 10 µm × 10 µm and also in the resistance from 1 to 10 kOhm of processed chips from the same as-grown graphene/SiC film.

The introduction of additional cleaning of the graphene surface in the chips after PLG resulted in RMS values close to the values on as-grown graphene/SiC film for the vast majority of the chips (up to 80%). As a result, the spread of chip resistance values decreased to 1–1.6 kOhm.

It has been shown that measurements of low-frequency noise are an effective tool for monitoring the non-reproducibility of graphene properties in a chip and finding out the causes of non-reproducibility. The spectral density of voltage fluctuations S_U_ > 10^−12^ V^2^/Hz at 1.22 Hz is typical for processed graphene with rather large local LRRs, clusters, or cracks or a high level of deformation. Moreover, the noise spectra S_U_(f) show the ranges of S_U_ ~1/f^γ^ (γ > 1), indicating the non-uniform deformations in graphene. At the same time, the value of γ allows us to compare the level of deformation in different chips and at different stages of graphene-based biosensor production. The revealed features of processed graphene with S_U_ > 10^−12^ V^2^/Hz lead to a deterioration in the adsorption properties of graphene, and the inhomogeneous attachment of biomolecules led to the deterioration of the detecting properties of the biosensor. In I vast majority of the chips after additional cleaning (R < 1.6 kOhm and S_U_ < 5 × 10^−13^ V^2^/Hz), regions with this type of dependence were not detected. Influenza viruses were successfully detected on chips with these parameters [10,11]. Thus, the assessment of the quality of processed graphene chips using AFM and measurements of low-frequency noise are useful tools for selecting and controlling the reproducibility of the parameters of chips intended for use in biosensors.

## Figures and Tables

**Figure 1 materials-16-05628-f001:**
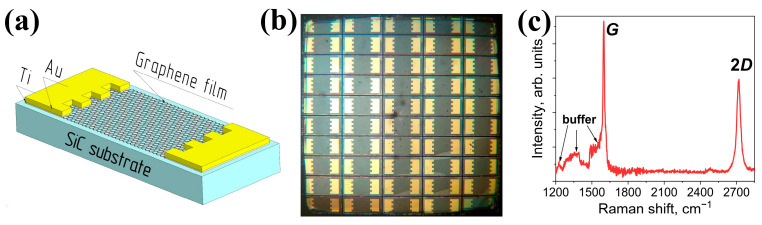
(**a**) Schematic illustration of a graphene chip on SiC substrate; (**b**) optical image of the as-grown graphene/SiC film after PLG, the size of the film is 11 mm × 11 mm; (**c**) Raman spectra of the as-grown graphene/SiC film before PLG.

**Figure 2 materials-16-05628-f002:**
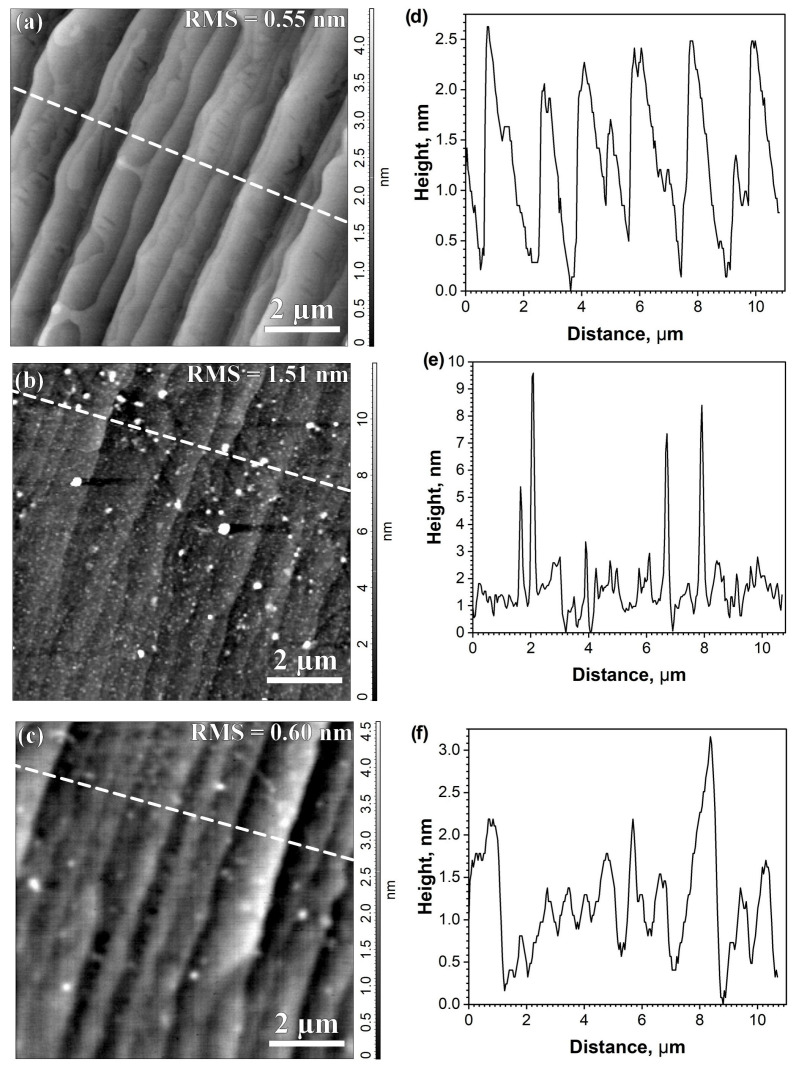
Graphene surface topography by the AFM on different stages of chip processing (scan 10 µm × 10 µm): (**a**) as-grown graphene/SiC surface (RMS = 0.55 nm); (**b**) the graphene surface in a chip without cleaning from LRRs (RMS = 1.51 nm); (**c**) the graphene surface in a chip with additionally cleaned LRRs (RMS = 0.60 nm); (**d**–**f**) the graphene surface profiles along the dotted line in a topographic image of the corresponding surface.

**Figure 3 materials-16-05628-f003:**
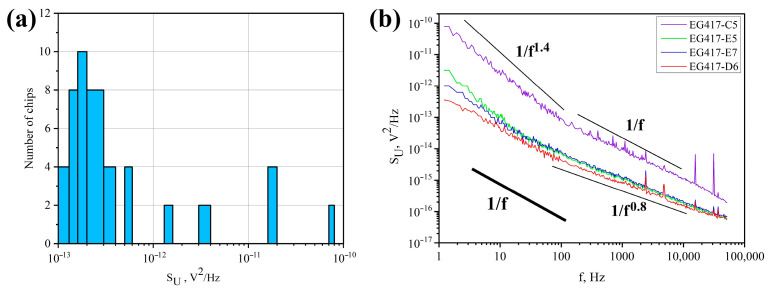
(**a**) Frequency dependence of the S_U_. The histogram of the S_U_ distribution at a frequency of 1.22 Hz for 50 chips obtained with additional cleaning of the LRRs; (**b**) low-frequency noise spectra of the chips with different S_U_ values. Black lines indicate a simulation of the 1/f^γ^ dependence with γ 0.8, 1, and 1.4 for references. The insert in Figure 3b indicates the chip numbers.

**Figure 4 materials-16-05628-f004:**
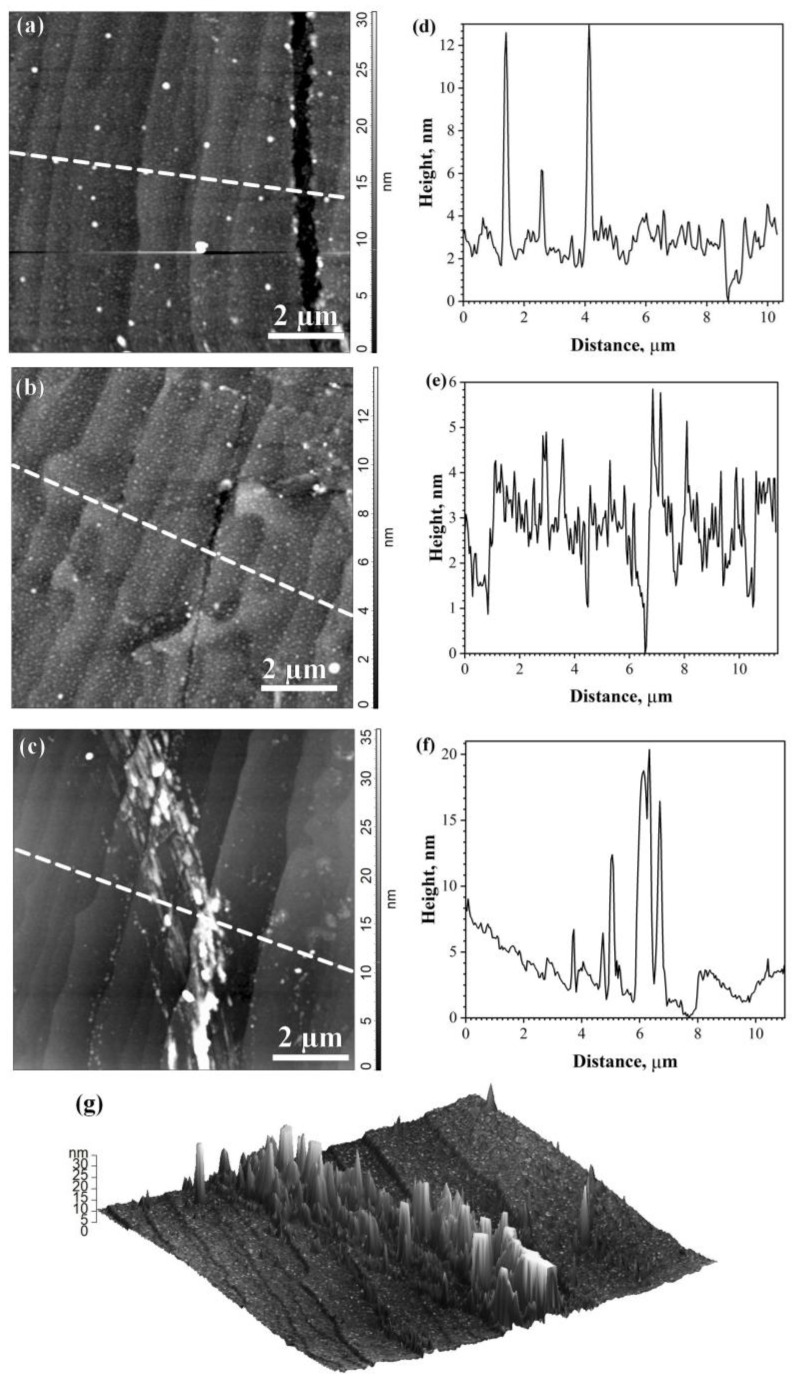
(**a**–**c**) Graphene surface topography in three chips with an excessive low-frequency noise S_U_ > 4 × 10^−13^ V^2^/Hz (scan 10 µm × 10 µm); (**d**–**f**) graphene surface profiles along the dotted line in a topographic image of the corresponding surface. (**g**) Three-dimensional image of graphene with LRRs.

## Data Availability

Data are contained within this article.

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
