# Peer review of "Quality Assessment of Processed Graphene Chips for Biosensor Application"

_materials, 2023, doi:10.3390/ma16165628_

Round 1

Reviewer 1 Report

Dear Authors,

Your manuscript should be improved with more information about the graphene, its usage in biosensors, advantages, disadvantages, quality parameters and related studies with the updated literature.

Your materials should be given with their commercial names, numbers etc.

You need to add references for the non-unique methods.  

Why did you choose only these experiments?

Results should be discussed in order to highlight scientific meanings. 

Discussion part should be included existing literature comparations.

Figure resolutions are not sufficient.

Minor editing of English language required.

Author Response

Response to Reviewer 1 Comments

Dear reviewer,

We deeply appreciate you for all your time spending on reviewing our manuscript and offering valuable comments and suggestions. All your comments and suggestions were taken into consideration in the revised manuscript. The revised parts are highlighted in red font in the revised manuscript For your convenience, below are our responses to your comments.

Point 1: “Your manuscript should be improved with more information about the graphene, its usage in biosensors, advantages, disadvantages, quality parameters and related studies with the updated literature.”

Response 1: The Introduction section has been largely rewritten (Lines 28-61, 78-97, 104-105, 117-121). Some unmodified parts of the text have changed the order for a better understanding of existing problems and ways to solve them (Lines 97-103, 105-117).

Point 2: “Your materials should be given with their commercial names, numbers etc.”

Response 2: The commercial names of the materials and equipment used have been added to the “Materials and Methods” section.

Point 3: “You need to add references for the non-unique methods. 

Response 3: Additional references to non-unique methods (AFM, Raman spectroscopy, determination of low-frequency noise characteristics) have been added to the section “Introduction“.

Lines 90-92: “We assess the quality of graphene in chips using traditional AFM [22] and Raman [23-25] studies and also the low-frequency noise (LFN) studies.”

Point 4: “Why did you choose only these experiments?”

Response 4: The choice of these experiments alone is determined both by our technical capabilities and by the belief that without preserving the properties of graphene in chips, at the level of properties of as-grown graphene film, it is difficult to develop effective and reproducible methods for attaching biomolecules on the graphene surface  and detecting them. The reasons for the non-reproducibility of graphene properties in chips identified by these experiments and the complex of control methods used made it possible to significantly improve the reproducibility of graphene properties in chips and allowed in subsequent experiments with biomolecules to visualize the attachment of antibodies and antigens to the graphene surface in chips.The following text was inserted into the manuscript on lines:

Lines 77-87: “One of the methods to resolve the problem is to use a protective (sacrificial) layer between graphene film and a photoresist to limit the interaction between them [19-21]. It should be noted that it is quite difficult to compare the effectiveness of the proposed meth-ods based on published data. The problem relates with the variety of graphene properties, obtained by different methods and also the lack of generally accepted graphene quality standards. Judging by the fact that since 2014 [19] and up to the present, works on the suc-cessful solution of the problem have been regularly published [11, 21], there does not seem to be a generally accepted effective way to solve the problem. Apparently, the easiest way is to compare the RMS surface roughness of the as-grown graphene film with RMS surface roughness of graphene in chips after PLG from the same film.”

Lines 226-228: “We chose this technique because SU values provide integral information about the state of the defect system of the whole gra-phene film in the chip, rather than some local areas.”

Point 5: “Results should be discussed in order to highlight scientific meanings”.

Response 5: The order of presentation in section “Results and Discussion” has been changed accordingly, as you indicated.

Lines 168-170: “To determine the reproducibility of graphene properties in chips processed from as-grown graphene/SiC films, the Raman spectra and surface topography of graphene were studied both in chips and in as-grown graphene/SiC films.”

Lines 183-209: “ The study and comparison of the properties of graphene in chips were carried out on chips obtained from the same film.

It is known that the processing of graphene chips from as-grown graphene/SiC films using conventional PLG with the removal of a photoresist in acetone leads to a deterioration in the morphology of the graphene surface and a set of LRRs [11, 17, 18]. Our experiments have shown that LRRs contribute to the non-reproducibility of chip parameters [11]. A comparison of the AFM images in Fig.2a and 2b reveals typical change in the surface topography of graphene caused by the formation of LRRs visible as white local regions in AFM images. The presence of LRRs results in a spread RMS values from 0.5 to 10 nm at AFM scan of 10 µm × 10 µm in graphene chips from the same film. At the same time, there was a spread of resistance values (R) in chips from 1 to 10 kΩ, which is associated with the spread of RMS values. Large values of resistance in chips, as a rule, were accompanied by a large RMS value of graphene surface roughness. As a result, only 20% of the 50 chips, which had resistance values in the range of 1-1.6 kΩ and RMS roughness from 0.5 to 1 nm with AFM scanning of 10 µm × 10 µm, can be considered as chips with near the same (uniform) parameters.”

In this work, to remove LRRs identified by AFM additional cleaning based on a piranha solution H2O2 (30%) + H2SO4 (concentrated) 1:2, (Neva Reaktiv Co., Ltd. Saint-Petersburg, Russia, https://nevareaktiv.ru/)  was used. After the cleaning, 50 chips fabricated from the as-grown graphene/SiC film(EG417 series) were evaluated based on the AFM characterization, the results of resistance measurement, the level of spectral density of a low-frequency noise (SU) at a frequency of 1.22 Hz, as well as analyzing the shape of SU frequency dependencies. No RMS values larger than 2 nm were observed in graphene in the vast majority of chips. The typical RMS values are about 0.60 nm (Fig. 2c), which coincides or is close to the initial RMS values of as-grown graphene/SiC film (RMS = 0.55 nm) Fig. 2a. A significant reduction in the spread of resistance values between the chips to 1-1.6 kΩ was observed.

Lines 210-217: It is difficult to say that the method used is better than the technique described earlier [19, 21], which uses the introduction of a sacrificial layer (an additional photoresists) or thin metal films. There are no generally accepted standards not only for the quality of the graphene surface after post-growth processes, but also for the as-grown graphene surface. That is why we compare the topography of the graphene surface and the RMS values after the conventional removal of the photoresist in acetone and after additional сleaning, as well as with the RMS values of graphene on the same as-grown film before PLG.

Lines 224-231: To find out the reasons for the resistance spread in the range 1-1.6 kΩ, measurements of the spectral density of voltage fluctuations SU at a frequency of 1.22Hz were carried out for all 50 chips from the same batch (EG417). We chose this technique because SU values provide integral information about the state of the defect system of the whole graphene film in the chip, rather than some local areas. The histogram of the distribution of SU values is shown in Figure. 3a. For 82% of chips, the SU values are as low as SU = (1-4)×10-13 V2/Hz that indicates a good quality graphene. Selective check-up of the Rаman spectra for these chips shows that they are identical to those shown in Fig. 1c

Lines 239-256: For the remaining 18% of chips, the spread of SU values was in the range from 6×10-13 to 8×10-11 V2/Hz (Figure 3a). Assuming that the observed variation of SU in these chips can be caused by inhomogeneous deformations in graphene, noise spectra SU(f) were investigated. It is known that the shape of noise spectra SU ~ 1/fγ (γ > 1) reveals the information about inhomogeneous deformations in the material [26, 27]. Indeed, for the chips EG417-C5, Е5, Е7 chips showing different values of excess noise, there are regions with frequency dependencies SU ~1/fγ having different values of γ up to 1.4 with a maximum SU value at 1.22 Hz for the EG417-C5 chip as shown in Fig.3b. We have earlier observed that the dependence SU ~ 1/fγ (γ >1) in graphene films and the chips is consistent with the Raman spectra data that demonstrate an inhomogeneous distribution of compressive stresses in graphene [32]. Studying of graphene topography in these chips reveals cracks in graphene shown in Figures 4a, b, c, and d (EG417-Е7, -C5) and extended LRRs regions shown in Figures 4e and g (EG417-E5) or a large number of heterogeneously spaced LRRs (Figures 2b and 4e). We clarify that the formation of cracks is observed, as a rule, in processed graphene in chips after PLG. Several causes increase the likelihood of crack formation: boiling in acetone, elevated temperatures of photoresist hardening, prolonged exposure to ultrasound, and local deformations of graphene, as well as the ripple of as-grown graphene film.

Point 6: “Discussion part should be included existing literature comparations.”

Response 6: In accordance with your comment, we have included in section “Results and Discussion” a discussion of the results obtained with the results of existing publications [19] and [21].

Lines 210-217: It is difficult to say that the method used is better than the technique described earlier [19, 21], which uses the introduction of a sacrificial layer (an additional photoresists) or thin metal films. There are no generally accepted standards not only for the quality of the graphene surface after post-growth processes, but also for the as-grown graphene surface. That is why we compare the topography of the graphene surface and the RMS values after the conventional removal of the photoresist in acetone and after additional сleaning, as well as with the RMS values of graphene on the same as-grown film before PLG.

Lines 282-285: The introduction of additional cleaning of the graphene surface in the chips also made it possible, as in [19,21], to prevent the formation of LRRs and to increase the reproducibility of graphene chip parameters.

Point 7: ”Figure resolutions are not sufficient.”

Response 7:  We improved figure reolution from 96 dpi to 300 dpi

Point 8: “Minor editing of English language required.”

Response 8: The English language was edited in terms of shortening and simplifying sentences for a better understanding of the text, as well as changing the presentation of the text.

Reviewer 2 Report

Residue in graphene is a very commend problem in two dimensional devices and has been extensively studied, especially with the characterization techniques of Raman and AFM. 

On the other hand, cleaning of residue of graphene, especially post device fabrication can still draw broad interests, where the authors did not put too much focus on. 

The manuscript can not be published at the current stage, while I'd like to re-consider after major changes made. I suggest the authors focus more on the residue cleaning method, what is the detail method to clean graphene, why does this cleaning method work, how this method is better than the current works.

No comments at this point. 

Author Response

Dear reviewer:

We deeply appreciate you for all your time spending on reviewing our manuscript and offering valuable comments and suggestions. All your comments and suggestions were taken into consideration. The revised parts are marked in red in the revised manuscript. For your convenience, below are our responses to your comments.

Point 1: “Residue in graphene is a very commend problem in two dimensional devices and has been extensively studied, especially with the characterization techniques of Raman and AFM.

On the other hand, cleaning of residue of graphene, especially post device fabrication can still draw broad interests, where the authors did not put too much focus on.

The manuscript can not be published at the current stage, while I'd like to re-consider after major changes made. I suggest the authors focus more on the residue cleaning method, what is the detail method to clean graphene, why does this cleaning method work, how this method is better than the current works.”

Response 1: We have made appropriate changes to all sections of the manuscript for a better understanding of the experiments and their results.

In particular, the following passages are additionally included in the text of the manuscript:

Lines 77-87: “One of the methods to resolve the problem is to use a protective (sacrificial) layer between graphene film and a photoresist to limit the interaction between them [19-21]. It should be noted that it is quite difficult to compare the effectiveness of the proposed meth-ods based on published data. The problem relates with the variety of graphene properties, obtained by different methods and also the lack of generally accepted graphene quality standards. Judging by the fact that since 2014 [19] and up to the present, works on the suc-cessful solution of the problem have been regularly published [11, 21], there does not seem to be a generally accepted effective way to solve the problem. Apparently, the easiest way is to compare the RMS surface roughness of the as-grown graphene film with RMS surface roughness of graphene in chips after PLG from the same film.”

Lines 199-217 “In this work, to remove LRRs identified by AFM additional cleaning based on a piranha solution H2O2 (30%) + H2SO4 (concentrated) 1:2, ((Neva Reaktiv Co., Ltd. Saint-Petersburg, Russia, https://nevareaktiv.ru/ ) was used. After the cleaning, 50 chips fabricated from the as-grown graphene/SiC film (EG417 series) were evaluated based on the AFM characterization, the results of resistance measurement, the level of spectral density of a low-frequency noise (SU) at a frequency of 1.22 Hz, as well as analyzing the shape of SU frequency dependencies. No RMS values larger than 2 nm were observed in graphene in the vast majority of chips. The typical RMS values are about 0.60 nm (Fig. 2c), which coincides or is close to the initial RMS values of as-grown graphene/SiC film (RMS = 0.55 nm) Fig. 2a. A significant reduction in the spread of resistance values between the chips to 1-1.6 kΩ was observed.

It is difficult to say that the method used is better than the technique described earlier [19, 21], which uses the introduction of a sacrificial layer (an additional photoresists) or thin metal films. There are no generally accepted standards not only for the quality of the graphene surface after post-growth processes, but also for the as-grown graphene surface. That is why we compare the topography of the graphene surface and the RMS values after the conventional removal of the photoresist in acetone and after additional сleaning, as well as with the RMS values of graphene on the same as-grown film before PLG.”

Line 282-285 The introduction of additional cleaning of the graphene surface in the chips also made it possible, as in [19,21], to prevent the formation of LRRs and to increase the reproducibility of graphene chip parameters

Point 2.: Moderate editing of English language required

Response 2: The English language was edited in terms of shortening and simplifying sentences for a better understanding of the text, as well as changing the presentation of the text.

Reviewer 3 Report

1. What is the main question addressed by the research?
The main question is a reliable assessment of possible defects arising during chip processing of graphene for biosensor applications.

2. Do you consider the topic original or relevant in the field? Does it address a specific gap in the field?
Yes, definitely. Device yield and operation depends very much on processing details.

3. What does it add to the subject area compared with other published material?
Noise measurement is an original approach.

4. What specific improvements should the authors consider regarding the methodology? What further controls should be considered?
Some idea about a specific biosensor example would be of importance.

5. Are the conclusions consistent with the evidence and arguments presented and do they address the main question posed?
Yes.

6. Are the references appropriate?
yes

This is an interesting and practically useful paper.

Some recommendations which will increase the paper value:

- From the title it is not clear that the investigations concern "processed" graphene. Please correct this.

- Please, explain the origin and mechanism of crack formation in graphene.

-

There are some unclear sentences which can be improved by improving the English. Here I give some examples, but a check through out the whole text is recommended.

Line 31, Lines 54-57, Line 200

Author Response

Dear  reviewer:

We deeply appreciate you for all your time spending on reviewing our manuscript and offering valuable comments and suggestions. All your comments and suggestions were taken into consideration. The revised parts are highlighted in red font in the revised manuscript. For your convenience, below are our responses to your comments.

Point 1:  Some idea about a specific biosensor example would be of importance.

Response 1: The following text has been added to the "Introduction" section to get some idea of a specific biosensor example

Lines 44-54: “The concept of a graphene based biosensor is based on the antigen-antibody immunoreaction on the graphene surface.

Briefly, the main stages of creating biosensors include the formation of a graphene film, the formation of chips with contact pads (graphene resistors or transistor) by photolithography (PLG), controlled treatment (functionalization) of the graphene surface in chips, immobilization (attachment) of biomolecules (antibodies), the implementation of antibody–antigen reaction [3-9]. Such a sophisticated biosensor fabrication technique is necessary to increase the selectivity and sensitivity of biosensors. Its use has made it possible to obtain biosensors for viral infection, such as A and B influenza and COVID19 [3,5,11-16].“

Point 2: From the title it is not clear that the investigations concern "processed" graphene. Please correct this.

Response 2: We have modified the title of the manuscript to the following:

“Quality assessment of processed graphene chips for biosensor application”.

Point 3: Please, explain the origin and mechanism of crack formation in graphene.

Response 3: The following text has been added to the "Introduction" section

Lines 252-256: “We clarify that the formation of cracks is observed, as a rule, in processed graphene in chips after PLG. Several causes increase the likelihood of crack formation: boiling in acetone, elevated temperatures of photoresist hardening, prolonged exposure to ultrasound, and local deformations of graphene, as well as the ripple of as-grown graphene film.”

Point 4: Moderate editing of English language required.

“There are some unclear sentences which can be improved by improving the English. Here I give some examples, but a check through out the whole text is recommended.

Line 31,  Lines 54-57, Line 200 “

Response 4: The English language was edited in terms of shortening and simplifying sentences for a better understanding of the text, as well as changing the presentation of the text.

In particular, the text on line 31 has been changed as follows.

Before: “Currently it is not clear to what extent is it possible to preserve the quality of pristine graphene films in the process of chips fabrication and obtaining biosensors based on it.”

After: This sentense was removed. The second sentence following the first one in the text has been modified as follows

Lines 61-63:  “The quality of graphene films can be disturbed alrady at the initial stage of the chip formation of a given topology by photolithography (РLG) [11,17,19].”

The text on lines 54-57 has been changed as follows

Before: “ In this work, we report on assess the quality and reproducibility of graphene parameters in chips at the initial stage of their production not only AFM and Raman studies but with low-frequency noise studies.”

After:

Lines 88-92  “ In this work, we report on the quality assessment of processed graphene chips, ob-tained by traditional PLG and PLG with additional LRRs cleaning and also on reproduci-bility of their parameters for biosensor applications. We assess the quality of graphene in chips using traditional AFM [22] and Raman [23-25] studies and also the low-frequency noise (LFN) studies.”

The text on line 200 has been changed as follows.

Before: “As it was shown previously [9], the heterogeneous distribution of deformations in the graphene chip used as a biosensor leads to non-uniform spread of the immobilized of an-tibodies.”

After:

Lines 257-259: “As it was shown previously [9], the heterogeneous distribution of deformations in the graphene chip leads to non-uniform immobilization of antibodies”,

Round 2

Reviewer 1 Report

Dear authors,

Thanks for your corrections.

Best regards,

Minor English editing is required before publishing. 

Author Response

Dear reviewer:

We are grateful to you for the re-editing of our manuscript. The English language was edited in terms of shortening and simplifying sentences for a better understanding of the text, as well as changing the presentation of the text.

Reviewer 2 Report

I was expecting an author response letter with more insightful and constructive support, such as additional experiment and discussion. While the author failed to provide any of these, so I insist my previous point: there is no innovation of this manuscript, and I cannot see any benefit of providing this method to the field. 

Author Response

Dear reviewer:

We are grateful to you for the re-editing of our manuscript. Your comment showed us that we have not fully disclosed the details of our experiments and the results obtained. We have added a new text marked in green to the re-revised manuscript and additional materials in Supplemental Materials, a new section.

In this manuscript, it was important for us to present the results on the effectiveness of using the LFN method to control the reproducibility of the properties of processed graphene in chips.

Below are our responses to your comment.

Comment 1: “I was expecting an author response letter with more insightful and constructive support, such as additional experiment and discussion. While the author failed to provide any of these, so I insist my previous point: there is no innovation of this manuscript, and I cannot see any benefit of providing this method to the field..”

Response 1: The following passages are additionally included in the text of the manuscript:

Section 1 "Introduction"

Lines 84 - 90: “It should be noted that several papers provide data that the introduction of a protective (sacrificial) layer was insufficient. It was possible to obtain a graphene surface free of LRRs only with additional annealing [19, 20]. Thus, despite the successes achieved in cleaning the graphene surface, a generally accepted solution has not yet been found. Meanwhile, this is an important step in obtaining biosensors with reproducible parameters. ”

Lines 93 - 96: “Additional LRRs cleaning is based on the use of piranha solution, which is widely used in the microelectronics and in silicon technology to remove organic contaminants, but in graphene PLG, according to our data, was not used.

Section 2 " Materials and Methods "

Lines 141 - 143: “Аdditional cleaning based on a piranha solution H2O2 (30%) + H2SO4 (concentrated) 1:2, (Neva Reaktiv Co., Ltd. Saint-Petersburg, Russia, https://nevareaktiv.ru/ ) at 25- 270 C was used”

Section 3" Results and Discussion "

Lines 289 - 302: “The introduction of additional cleaning of the graphene surface in the chips also made it possible, as in [19,21], to preserve the quality of the graphene surface in chips after PLG with RMS values close to the RMS values as-grown graphene film before PLG, reduce the resistance of graphene in chips, increase the reproducibility of graphene resistance values in chips obtained from a single film, but in a simpler way than in [19, 21]. The quality assessment of processed graphene in chips using LFN method allowed us to control the heterogeneous distribution of deformations and isolate chips with cracked graphene that are not suitable for use in biosensors by the level of excessive noise. The use of this method in combination with AFM opens up opportunities to increase the reproducibility of the properties of processed graphene in chips, as well as biosensors, and to improve the technology for obtaining biosensors. The additional cleaning contributed to an increase in the sensitivity of viral biosensors and made it possible to successfully detect influenza B viruses (See more detals in Supplementary Materials and [11]).

We believe that the results presented in the manuscript have the following novelty:

  • We applied additional cleaning of the graphene surface in the chips using a piranha solution that had not previously been used to remove LRR from the graphene surface.
  • It is shown that in a simpler way than in [19, 21], the RMS values of graphene in chips are obtained as close as possible to the RMS values in as-grown graphene before PLG. In addition, resistance of the processed graphene chips has been reduced, as well as the spread of graphene chip resistance values. Data on the spread of resistance values are not given in the referred publications.
  • The possibilities of the LFN method for selecting graphene chips with non-reproducible integral properties of a defective system caused by inhomogeneous deformations of graphene and the formation of cracks are shown

We would like to emphasize the uniqueness of using the LFN method to assess the quality of the graphene surface for its application in biosensors.

The LFN method [26] integrally characterizes the system of defects of the entire area between the contacts in the chip, not only micro region as AFM and Raman methods. The manuscript shows that it is possible in a simpler way than the introduction of an sacrificial layer to obtain the surface quality of graphene in processed chips close to the original surface of the as-grown graphene. For graphene films, the dependence of the spectral density of a low-frequency current noise (SI) as well as that of a low-frequency voltage fluctuations (SU) on frequency have the form SU (or SI) ~ 1/fγ in the frequency range 1-100Hz. The quality of graphene and the reproducibility of its properties in the chips in general can be judged by comparing the magnitude of low-frequency noise at a frequency of 1.22 Hz between the chips with the same geometry and from the same film.